# Picropodophyllotoxin, an Epimer of Podophyllotoxin, Causes Apoptosis of Human Esophageal Squamous Cell Carcinoma Cells Through ROS-Mediated JNK/P38 MAPK Pathways

**DOI:** 10.3390/ijms21134640

**Published:** 2020-06-30

**Authors:** Ah-Won Kwak, Goo Yoon, Mee-Hyun Lee, Seung-Sik Cho, Jung-Hyun Shim, Jung-Il Chae

**Affiliations:** 1Department of Pharmacy, College of Pharmacy, Mokpo National University, Jeonnam 58554, Korea; rhkrdkdnjs12@mokpo.ac.kr (A.-W.K.); gyoon@mokpo.ac.kr (G.Y.); sscho@mokpo.ac.kr (S.-S.C.); 2College of Korean Medicine, Dongshin University, Naju, Jeollanam 58245, Korea; mhyun_lee@hanmail.net; 3Department of Dental Pharmacology, School of Dentistry and Institute of Oral Bioscience, BK21 Plus, Jeonbuk National University, Jeonju 54896, Korea

**Keywords:** picropodophyllotoxin, esophageal squamous cancer, apoptosis, p38, JNK

## Abstract

Esophageal squamous cell carcinoma (ESCC), a major histologic type of esophageal cancer, is one of the frequent causes of cancer-related death worldwide. Picropodophyllotoxin (PPT) is the main component of *Podophyllum hexandrum* root with antitumor activity via apoptosis-mediated mechanisms in several cancer cells. However, the underlying mechanism of the PPT effects in apoptosis induction in cancer remains ambiguous. Hence, in this study, we evaluate the anti-cancer effects of PPT in apoptotic signaling pathway-related mechanisms in ESCC cells. First, to verify the effect of PPT on ESCC cell viability, we employed an MTT assay. PPT inhibited the viability of ESCC cells in time- and dose-dependent manners. PPT induced G2/M phase cell cycle arrest and annexin V-stained cell apoptosis through the activation of the c-Jun N-terminal kinase (JNK)/p38 pathways. Furthermore, the treatment of KYSE 30 and KYSE 450 ESCC cells with PPT induced apoptosis involving the regulation of endoplasmic reticulum stress- and apoptosis-related proteins by reactive oxygen species (ROS) generation, the loss of mitochondrial membrane potential, and multi-caspase activation. In conclusion, our results indicate that the apoptotic effect of PPT on ESCC cells has the potential to become a new anti-cancer drug by increasing ROS levels and inducing the JNK/p38 signaling pathways.

## 1. Introduction

An estimated 18,440 new cases and 16,170 deaths from esophageal cancer are projected in the United States in 2020 [1,2,3,4,5]. Esophageal squamous cell carcinoma (ESCC), accounting for about 90% of the histologic types of esophageal cancer, is caused by alcohol consumption and tobacco use and genetic factors [3,4,6,7]. There are many approaches to treating ESCC, including endoscopic therapy, surgery, radiotherapy, chemotherapy, and adjuvant chemotherapy [2,6]. However, the mortality rate of ESCC patients has marginally improved over the last decades [2] and the survival rate is still low, due to recurrence or metastasis [1]. Furthermore, there are no specific ESCC chemotherapy drugs and even conventional anticancer drugs show resistance and side effects [8]. Therefore, it is necessary to develop ESCC anticancer agents from natural substances with relatively low side effects and drug resistance.

Picropodophyllotoxin (PPT) is an epimer of podophyllotoxin isolated from the roots of *Podophyllum hexandrum*, which is used as an anticancer drug and insecticidal/antifungal agent [9,10]. It has also been applied to remedy genital warts by inhibiting microtubule assembly [11]. Recent studies have shown that PPT inhibits microtubule assembly, resulting in antitumor effects without cytotoxicity [11,12]. Additionally, the cyclolignan PPT, one of the aryl tetralin lignans, was launched as an anticancer drug specifically targeting insulin-like growth factor 1 receptor (IGF-1R) [11]. It also induces G2/M phase arrest and apoptosis in multiple myeloma cells by inhibiting IGF-1R [13]. These findings suggest that PPT can be used as a natural anticancer drug. When IGF-1R, a transmembrane receptor tyrosine kinase, is activated, it promotes multiple downstream signaling including mitogen-activated protein kinase (MAPK) signaling pathways, which modulate the growth and survival of cancer cells [14]. The MAPK pathway is involved in the development and progression of cancer [15] and transforms extracellular stimuli into a broad range of cellular responses [15,16]. The c-Jun N-terminal kinase (JNK) and p38-MAPK pathways are activated by stresses, such as oxidative stress, cytokines, and ultraviolet radiation [17]. According to several studies, JNK/p38 MAPK signaling pathways activation suppressed tumorigenesis and cancer growth by promoting cell death [18,19,20]. However, it is not clear whether JNK and p38-MAPK signaling are involved in PPT-induced cancer cell apoptosis.

Therefore, we investigate whether PPT can induce apoptosis through the JNK/p38 MAPK pathways in ESCC cells. We report here that PPT induces apoptosis in ESCC cells by reactive oxygen species (ROS) generation, endoplasmic reticulum (ER) stress production, and JNK/p38 activation.

## 2. Results

### 2.1. PPT Inhibits ESCC Cell Proliferation and Anchorage-Independent Growth

The effects of PPT on ESCC cell viability were investigated by an 3-(4,5-dimethylthiazol-2-yl)-2,5 diphenyltetrazolium bromide (MTT) assay. KYSE 30, KYSE 70, KYSE 410, KYSE 450, and KYSE 510 cells were exposed to increasing concentrations (0.2, 0.3, and 0.4 µM) of PPT or dimethyl sulfoxide (DMSO) for 24 h and 48 h. As shown in Figure 1B–F, time-and dose-dependent decreases in cell viability were observed in all treated ESCC cells. The mean half-maximal inhibitory concentration (IC_50_) PPT values indicated a cytotoxic effect toward all ESCC cells (KYSE 30, 0.15 µM; KYSE 70, 0.32 µM; KYSE 410, 0.15 µM; KYSE 450, 0.26 µM; and KYSE 510, 0.24 µM). We used KYSE 30 and KYSE 450 cells for further experiments because both cells have a similar genetic background [21]. The soft agar assay results showed a similar decrease in cell growth capacity by PPT treatment, which indicated that the anchorage-independent growth of KYSE 30 and KYSE 450 cells was inhibited by PPT treatment (Figure 1G,H).

### 2.2. PPT Arrests G2/M Phase Cell Cycle Progression in ESCC Cells

We assessed the effects of PPT on cell cycle progression using a Muse™ Cell Analyzer (Merck Millipore, Darmstadt, Germany), since PPT inhibited ESCC cell viability. The cell cycle distribution of PPT-treated KYSE 30 and KYSE 450 cells showed increased G2/M phase accumulation (Figure 2A). The sub-G1 population of PPT-treated cells was significantly increased compared to DMSO-treated controls (Figure 2B). Accordingly, we examined the molecular mechanism of PPT-induced cell cycle arrest in ESCC cells by using Western blots (Figure 2C). The expression of p21 and p27 proteins, G2/M phase cell cycle regulators, significantly increased, whereas the levels of cyclin B1 and cdc2 proteins, cell cycle promoters, decreased in a dose-dependent manner (Figure 2C). These results suggest that PPT induced the G2/M phase arrest of ESCC cells.

### 2.3. PPT Induces Apoptotic Death of ESCC Cells

To identify whether PPT inhibited cell proliferation through apoptosis, we performed an annexin V/7-Aminoactinomycin D (7-AAD) apoptosis detection assay (Figure 3A). The right bottom and right upper panel of the dots in the plot indicate apoptotic cells stained with annexin V or 7-AAD. Treatment of the KYSE 30 and KYSE 450 cells with various doses of PPT (0.2, 0.3, and 0.4 µM) or DMSO for 48 h resulted in significant increases in the number of total apoptotic cells, while the percentage of viable cells decreased. The total cell apoptosis rate of the KYSE 30 cells was 5.78 ± 0.48% (DMSO), 24.60 ± 2.44% (0.2 µM PPT), 55.88 ± 1.44% (0.4 µM PPT), and 70.50 ± 2.32% (0.4 µM PPT). Similar to the KYSE 30 cells, the total cell apoptosis rate of the KYSE450 cells was 4.22 ± 0.29% (DMSO), 16.85 ± 1.11% (0.2 µM PPT), 43.78 ± 2.13% (0.3 µM PPT), and 72.76 ± 0.62% (0.4 µM PPT) (Figure 3A). Next, we examined whether JNK/p38 MAPKs were involved in the PPT-induced apoptosis in ESCC cells. As shown in Figure 3B, PPT significantly induced the phosphorylation of JNK and p38 proteins in ESCC cells in a dose-dependent manner. However, the total expression of both proteins was not changed. Thus, PPT induced apoptosis by activating JNK/p38 in ESCC cells.

### 2.4. PPT Induces the Generation of ROS and ER Stress in ESCC Cells

Previously, we reported that natural product-induced apoptosis triggered ROS generation and ER stress in cancer cells [22]. To confirm ROS generation by PPT treatment, we accessed the intracellular ROS levels by staining KYSE 30 and KYSE 450 cells using the Muse™ ROS reagent kit. As shown in Figure 4A, we found that PPT dose-dependently increased the generation of intracellular ROS in the ESCC cells. Upregulated ROS levels induce ER stress and increase the levels of unfolded ER-related protein [23,24]. C/EBP homologous protein (CHOP) and 78-kDa glucose-regulated protein (GRP78) have been regarded as two essential proteins in the ER stress response [22,25]. To confirm whether ER stress was induced by PPT treatment in the ESCC cells, we analyzed the expression of GRP78 and CHOP. The Western blot analysis results demonstrated the concentration-dependent induction of GRP78 and CHOP in response to PPT-treatment (Figure 4B). Since CHOP is associated with the expression of death receptors (DRs) [26], we confirmed that PPT influenced the levels of DR4 and DR5 protein. As shown in Figure 4B, PPT treatment increased the levels of DR4 and DR5 protein in a dose-dependent manner. Therefore, PPT treatment led to apoptosis by ROS generation and ER stress in the ESCC cells.

### 2.5. PPT Affects MMP and Regulates Mitochondria Apoptosis-Associated Protein in ESCC Cells

Changes in mitochondrial membrane potential are considered critical signals in mitochondria-mediated cell apoptosis [27,28]. Therefore, we analyzed mitochondrial membrane potential (MMP) by staining PPT-treated ESCC cells with Muse™ MitoPotential assay reagents (Merck Millipore, Darmstadt, Germany). The results showed that 0.4 µM PPT led to the depolarization of the MMP of 46.74 ± 1.59% and 44.69 ± 3.29% in the KYSE 30 and KYSE 450 ESCC cells, respectively (Figure 5A). Then, we confirmed mitochondria-mediated apoptosis and its related biomarkers (Figure 5B). The Western blot results revealed that PPT decreased the expression of Bid, Mcl-1, and Bcl-2, and increased the expression of Bax, apoptotic protease activating factor-1(Apaf-1), and cleaved Poly (ADP-Ribose) Polymerase (c-PARP) in the ESCC cells. In addition, the level of cytochrome C (cyto C) in mitochondria decreased remarkably, which differed from the increase in cyto C in the cytosol (Figure 5B). These data imply that PPT induced MMP dysfunction and mitochondria-mediated apoptosis.

### 2.6. PPT Promotes ESCC Cells Apoptosis Through Increasing Multi-Caspase Activity

Caspase, a pro-apoptotic protein, is activated predominantly by the programmed apoptosis pathway [29]. Therefore, we examined multi-caspase activity (caspase-1, -3, -4, -5, -6, -7, -8, and -9) to verify the apoptotic mechanism of PPT in the ESCC cells. At 0.4 µM of PPT, the total multi-caspase activity of KYSE 30 cells significantly increased to 74.96 ± 1.52%, more than the control cells of 5.76 ± 0.20% (Figure 6). Furthermore, the total multi-caspase activation of the KYSE 450 cells by PPT significantly increased at concentrations of 0.2, 0.3, and 0.4 µM with activities 12.95 ± 0.38%, 28.55 ± 0.61%, and 46.57 ± 1.48%, respectively, higher than the control cells at 4.05 ± 0.20% (Figure 6). These results suggest that PPT led to the activation of multi-caspases in KYSE 30 and KYSE 450 cells in a dose-dependent manner.

## 3. Discussion

PPT is one of well-known naturally occurring aryltetralin lignans revealing anti-tumor activity [30]. Although PPT has indicated anti-tumor effects in several cancers [11,12], the mechanism is, as yet, not clear. In this study, we explored the activation of JNK/p38 and ROS- and/or mitochondria-mediated caspases as potential mechanisms for PPT-induced apoptosis in ESCC cells. The present study used a cellular model in which apoptosis was analyzed to examine the anti-cancer effects of PPT on ESCC esophageal cancer growth. We demonstrated that PPT suppressed the cell viability (Figure 1B–F) and colony formation (Figure 1G) of the ESCC cells in time- or dose-dependent manners in vitro. According to previous reports, PPT has also been demonstrated to be nontoxic in rodents (Lethal Dose 50 ≥ 500 mg/kg) [31]. Therefore, PPT can effectively inhibit ESCC cells, and further studies on the inhibitory effects are needed. Many anti-cancer agents can decrease cancer cell viability by inducing ROS- and JNK/p38 MAPK-mediated apoptosis and arresting the cell cycle at the G2/M phase [32,33,34]. Cell cycle analysis following the treatment of ESCC cells with different concentrations of PPT showed higher numbers of cells in the G2/M phase compared to the DMSO-treated control cells (Figure 2A). These results demonstrate that PPT inhibited cell viability by G2/M phase arrest in a dose-dependent manner. The cell cycle arrest at the G2/M phase resulting from PPT treatment led us to determine the expression level of cell cycle-regulated proteins. Cdc2 is a general regulator of G2/M transition in the eukaryotic cell cycle and can be activated by conjoining with cyclin B1 to initiate the M phase of the cell cycle [35]. p21 and p27, members of the Cip/Kip family, inhibit cyclin-dependent kinase complexes [36]. PPT-induced cell cycle retardation at the G2/M phase by increasing the expression of p21 and p27 and decreasing the levels of cyclin B1 and cdc2 (Figure 2B). Apoptosis, type I cell death among other physiologically distinct cell deaths, is divided into two major pathways: the intrinsic (the mitochondrial) pathway and the extrinsic (the death receptor) pathway [29]. JNK/p38 MAPK can associate with apoptosis in response to extracellular stimuli, such as chemotherapeutic agents [15]. In this study, we revealed the PPT-induced phosphorylation of JNK/p38 MAPK by Western blotting after the ESCC cells were treated with PPT for 48 h. Higher levels of ROS in cells are reported to induce cell cycle arrest, apoptosis, or senescence [33,37]. Under ER stress, the activation of GRPs, such as GRP78 and GRP94, has been shown to regulate toxicant- or stimulus-induced apoptotic pathways [38,39]. Furthermore, the expression of CHOP in the nucleus is upregulated during apoptosis induced by ER stress [22,27,38,39]. ER stress facilitates the induction of DRs that may be invoked in apoptosis [40]. In this study, PPT upregulated ROS generation (Figure 4A) and ER stress-mediated protein expression (Figure 4B). These results suggest that the ROS- and ER stress-mediated apoptotic pathway may play a crucial role in PPT-induced apoptosis. The increased generation of ROS and ER stress causes a loss in MMP, resulting in the release of cyto C from the membrane to the cytosol and this induces caspase-dependent signaling [23,25,29,37,40]. Our study results showed that PPT induced MMP dysfunction (Figure 5A) and apoptosis via regulation of mitochondria-associated apoptotic protein (Figure 5B). We also demonstrated that treatment with PPT activated multiple caspases in ESCC cells (Figure 6).

In conclusion, the results of our study revealed that PPT treatment was able to effectively induce apoptosis by regulating anti- and pro-apoptotic proteins in human ESCC cells via the JNK/p38 MAPK signaling pathways, which demonstrated the detailed anti-cancer mechanism of PPT in human ESCC cells. Combination therapy, which combines two or more treatments, is currently a central component of cancer therapy [41]. Fusion of anti-cancer drugs ameliorates efficacy compared to mono-therapy and potentially decreases drug resistance, offering therapeutic anti-cancer benefits such as reducing tumor growth and metastatic potential, cell cycle arrest, and induction of apoptosis [41,42]. The findings of this study suggest that PPT may be valuable as a potential anti-cancer agent and be considered valuable candidates of combination therapy for human ESCC.

## 4. Materials and Methods

### 4.1. Chemicals and Reagents

RPMI-1640, fetal bovine serum (FBS), phosphate-buffered saline (PBS), penicillin, streptomycin, and trypsin were obtained from Hyclone (Logan, UT, USA). Primary antibodies against actin, p21, p27, cyclin B1, cdc2, Bax, Bcl-2, Bid, Mcl-1, GRP78, CHOP, DR4, DR5, Bid, cyto C, α-tubulin, COX4, Apaf-1, and PARP were purchased from Santa Cruz Biotechnology (Santa Cruz, CA, USA). Phosphor (p)-JNK, JNK, p-p38, and p38 antibodies were supplied by Cell Signaling Technology (Danvers, MA, USA). DMSO, MTT, and PPT were purchased from Sigma-Aldrich, Inc. (St. Louis, MO, USA).

### 4.2. Cell Culture

Human ESCC cell lines, KYSE 30, KYSE 70, KYSE 410, KYSE 450, and KYSE 510 were provided by the Type Culture Collection of the Chinese Academy of Sciences (Shanghai, China). The cells were cultured in RPMI-1640 medium containing 10% FBS, and 100 U/mL penicillin and streptomycin at 37 °C in a humidified incubator with 5% CO_2_.

### 4.3. Cell Viability Assay

The proliferation activity of the ESCC cells was measured by an MTT assay. The cells (KYSE 30 (2.75 × 10^3^/well), KYSE 70 (10 × 10^3^/well), KYSE 410 (2.5 × 10^3^/well), KYSE 450 (3.5 × 10^3^/well), and KYSE 510 (5.5 × 10^3^/well)) were seeded into 96-well plates with RPMI-1640 medium containing 10% FBS and then incubated for 24 h or 48 h in the presence of different concentrations of PPT. MTT solution (5 mg/mL, 30 µL) was added to each well and the plates were incubated at 37 °C for 1 h and the supernatant was removed. After dissolving the formazan crystals in 100 µL of DMSO for five min, the absorbance of each well was measured using a spectrophotometer (Thermo Fisher Scientific, Vantaa, Finland) at 570 nm.

### 4.4. Soft Agar Assay

KYSE 30 and KYSE 450 (8 × 10^3^/well) cells were suspended in 0.3% agar with BME medium containing FBS, L-glutamine, and gentamicin containing DMSO or PPT (0.2, 0.3 and 0.4 µM) and layered on the top of wells in a 6-well plate with a base layer of 0.6% agar in BME medium containing FBS, L-glutamine, and gentamicin with DMSO or PPT (0.2, 0.3 and 0.4 µM). The cells were maintained at 37 °C in a 5% CO_2_ incubator for two weeks and the colonies were counted and photographed under light microscopy (Leica Microsystems, Wetzlar, Germany).

### 4.5. Cell Cycle Analysis

The KYSE 30 and KYSE 450 cells were cultured for 24 h in 6-well plates and treated with different concentrations of PPT for 48 h. The cells were washed with PBS and fixed with 70% ice-cold ethanol overnight. The cells were stained with Muse™ Cell Cycle reagent (Merck Millipore, Darmstadt, Germany) at 37 °C in the dark for 30 min. The cell cycle distribution was analyzed by using a Muse™ Cell Analyzer (Merck Millipore, Darmstadt, Germany).

### 4.6. Western Blots

Proteins were extracted from the ESCC cells treated with DMSO or PPT by lysing with radioimmunoprecipitation assay buffer and sonicating. The protein concentrations were analyzed using the DC Protein Assay (Bio-RAD, Hercules, CA, USA). Equal proteins were loaded onto sodium dodecyl sulfate-polyacrylamide gels, electrophoresed, and transferred to a polyvinylidene fluoride membrane (Merck Millipore, Bedford, UK). The membrane was incubated with primary antibody and washed six times with PBS containing 0.01% Tween 20 for 5 min. After washing, the membrane was incubated with the appropriate secondary antibody. The membrane was visualized with chemiluminescent detection reagents using ImageQuant LAS 500 (GE Healthcare, Uppsala, Sweden).

### 4.7. Annexin V/7-Aminoactinomycin D (7-AAD) Staining

The apoptosis rate of the cells was confirmed by using a Muse™ Annexin V & Dead Cell kit (Merck Millipore) according to the manufacturer’s protocols. In brief, the cells were washed with PBS and resuspended in 100 µL of Muse™ Annexin V & Dead Cell reagent (Merck Millipore) for 20 min at room temperature (RT) in the dark. The dead cells were analyzed using a Muse™ Cell Analyzer (Merck Millipore).

### 4.8. ROS Assay

The KYSE 30 and KYSE 450 cells were plated at a density of 7.5 × 10^4^/well and 10.5 × 10^4^/well, respectively, in 6-well plates. The cells were suspended in 190 µL of Muse™ Oxidative Stress Reagent working solution (Merck Millipore) and the fluorescent intensity was observed using a Muse™ Cell Analyzer (Merck Millipore).

### 4.9. Mitochondrial Membrane Potential Assay

The changes in MMP were analyzed using a Muse™ MitoPotential kit (Merck Millipore). KYSE 30 and KYSE 450 cells were exposed to PPT for 48 h and stained with Muse™ MitoPotential working solution at 37 °C for 20 min, followed by the addition of 5 µL of aminoactinomycin D (7-AAD). The samples were measured on a Muse™ Cell Analyzer (Merck Millipore).

### 4.10. Preparation of Cytosolic and Mitochondrial Fractions

The cytosolic fraction was extracted by suspending the PPT-treated cells with plasma membrane extraction buffer [250 mM sucrose, 10 mM HEPES (pH 8.0), 10 mM KCl, 1.5 mM MgCl_2_·6H_2_O, 1 mM EDTA, 1 mM EGTA, 0.1 mM phenylmethylsulfonyl fluoride, 0.01 mg/mL aprotinin, and 0.01 mg/mL leupeptin] containing 0.05% digitonin, and centrifuging at 13,000 rpm for 30 min. The pellets consisting of mitochondria were resuspended with plasma membrane extraction buffer and 0.5% Triton X-100 was added, followed by incubation on ice for 10 min. The mitochondrial fraction was collected by centrifugation at 13,000 rpm for 30 min.

### 4.11. Multi-Caspase Activity

Multi-caspase (caspase-1, -3, -4, -5, -6, -7, -8, and -9) activity was analyzed on the basis of the cleavage of caspase substrate using a Muse™ Multi-Caspase kit from Merck Millipore. The analyses were conducted according to the protocol provided by the manufacturer. The cells were exposed to Muse™ Multi-Caspase Reagent working solution for 30 min at 37 °C and Muse™ Caspase 7-AAD working solution was added, followed by incubation for five min at RT in the dark. The stained samples were analyzed using a Muse™ Cell Analyzer (Merck Millipore).

### 4.12. Statistical Analysis

The data are presented as means ± standard deviation (SD). Statistical analysis of the data was performed using the Prism 5.0 statistical package. The statistical significance of the differences between groups was analyzed using ANOVA and Fisher’s least significant difference posthoc test. The mean values were considered statistically significant at *p* < 0.05.

## Figures and Tables

**Figure 1 ijms-21-04640-f001:**
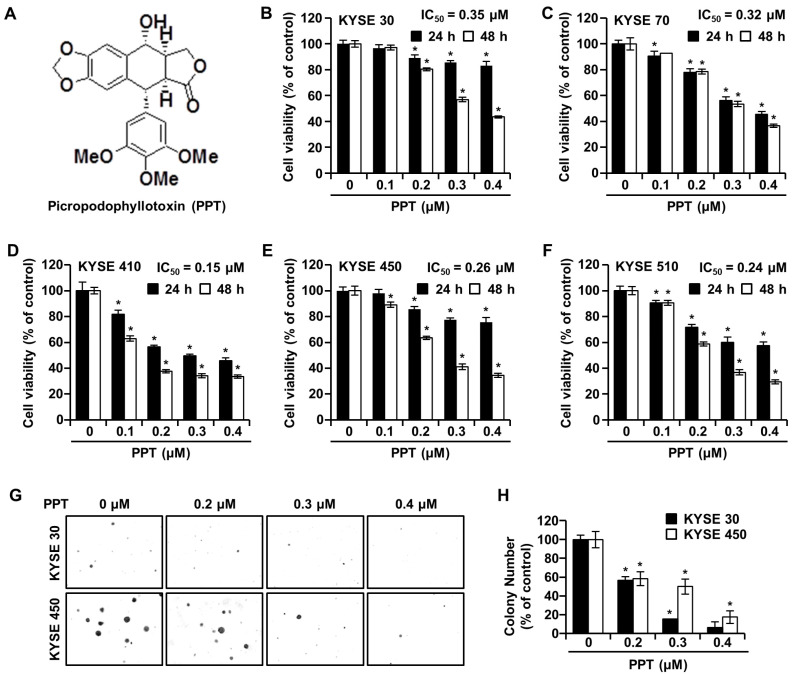
Picropodophyllotoxin (PPT) inhibits cellular viability and decreases colony formation in esophageal squamous cell carcinoma (ESCC) cells. (**A**) The chemical structure of PPT. (**B**–**F**) The cell viability of KYSE 30, KYSE 70, KYSE 410, KYSE 450, and KYSE 510 cells was measured following 24 h and 48 h exposure to 0.1, 0.2, 0.3, and 0.4 µM of PPT or dimethyl sulfoxide (DMSO) using an MTT assay. Each bar indicates the mean ± standard deviation (SD) of three independent experiments. * *p* < 0.05. (**G**) A soft agar assay of KYSE 30 and KYSE 450 cells was used to confirm colony growth and the long-term effects of PPT (0.2, 0.3, and 0.4 µM) compared to DMSO treatment. (**H**) Colony number results from soft agar analysis. The error bars represent the mean ± SD (n = 3, and * *p* < 0.05 vs. control).

**Figure 2 ijms-21-04640-f002:**
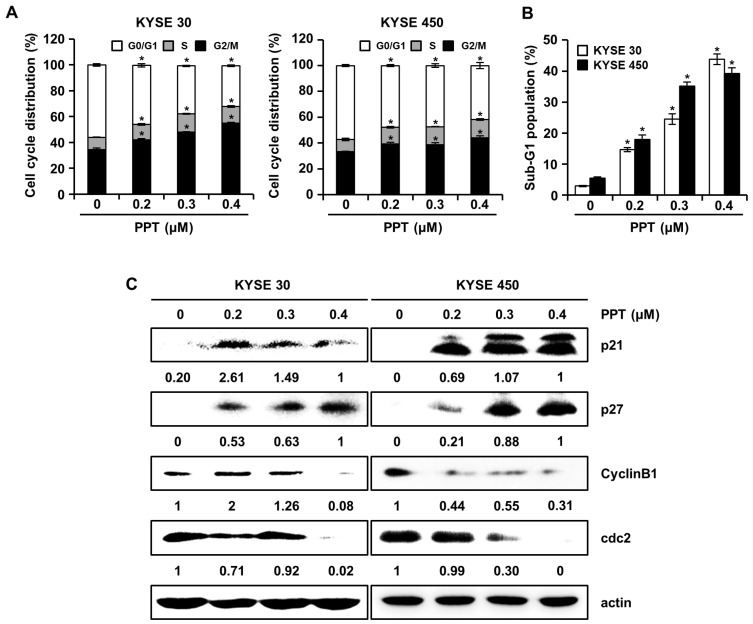
PPT causes cell cycle arrest at the G2/M phase in ESCC cells. KYSE 30 and KYSE 450 cells were treated with vehicle or 0.2, 0.3, and 0.4 µM PPT for 48 h. (**A**) The cells were stained with propidium iodide (PI) and cell cycle distribution was analyzed by a Muse™ Cell Analyzer (Merck Millipore, Darmstadt, Germany). Data present the mean ± SD of triplicate independent experiments; * *p* < 0.05, compared to the control cells. (B) The percentage of cells in the sub-G1 phase in the KYSE 30 and KYSE 450 cells is graphed. Each experiment was performed three times. The values are graphed as the means ± SD of three independent experiments for each treatment (* *p* < 0.05 compared to untreated controls). (**C**) The expression of p21, p27, cyclin B1, and cdc2 protein in DMSO or PPT-treated ESCC cells was analyzed by Western blots. Actin was used as a loading control.

**Figure 3 ijms-21-04640-f003:**
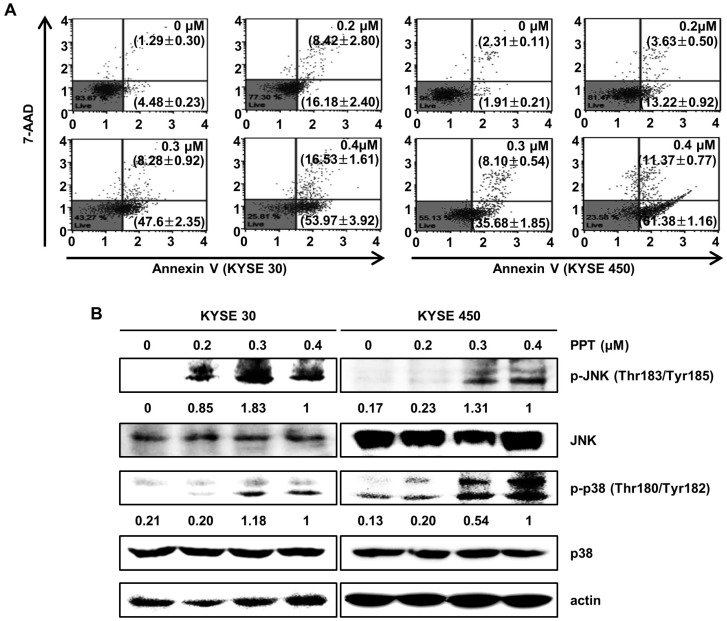
PPT induces cell apoptosis in a dose-dependent manner through the c-Jun N-terminal kinase (JNK)/p38 signaling pathways. KYSE 30 and KYSE 450 cells were treated with different concentrations of PPT (0.2, 0.3, and 0.4 µM) or DMSO for 48 h. (**A**) The cells were double-stained with annexin V/7-aminoactinomycin D (7-AAD). The apoptotic effects of PPT were assessed using a Muse™ Cell Analyzer (Merck Millipore) in the ESCC cell lines compared to the DMSO-treated control group. (**B**) The level of apoptosis-related protein phosphor (p)-JNK and p-p38 in the ESCC cells was detected by Western blot assays with actin as a loading control.

**Figure 4 ijms-21-04640-f004:**
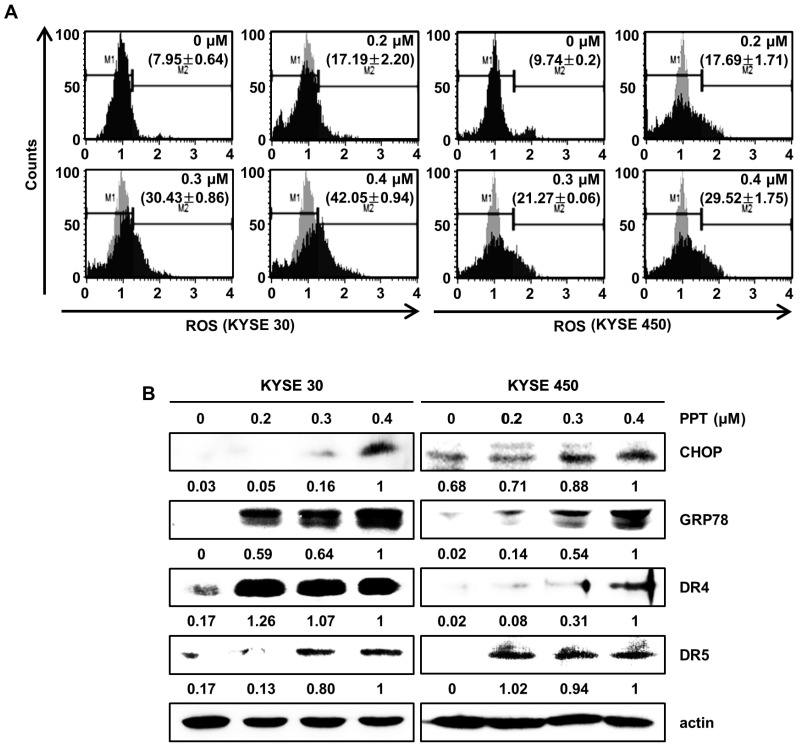
PPT provokes apoptosis by the generation of intracellular reactive oxygen species (ROS) and the regulation of endoplasmic reticulum (ER) stress-related proteins. ESCC cells (KYSE 30 and KYSE 450) were treated with PPT of 0.2, 0.3 and 0.4 µM or DMSO for 48 h. (**A**) Intracellular ROS was assayed using fluorescent dyes and Muse™ Oxidative Stress Reagent (Merck Millipore); n = 4. (**B**) The expression of ER stress-related proteins, including GRP78, C/EBP homologous protein (CHOP), DR4, and DR5, in PPT-treated ESCC cells was analyzed by Western blots. Actin served as a loading control.

**Figure 5 ijms-21-04640-f005:**
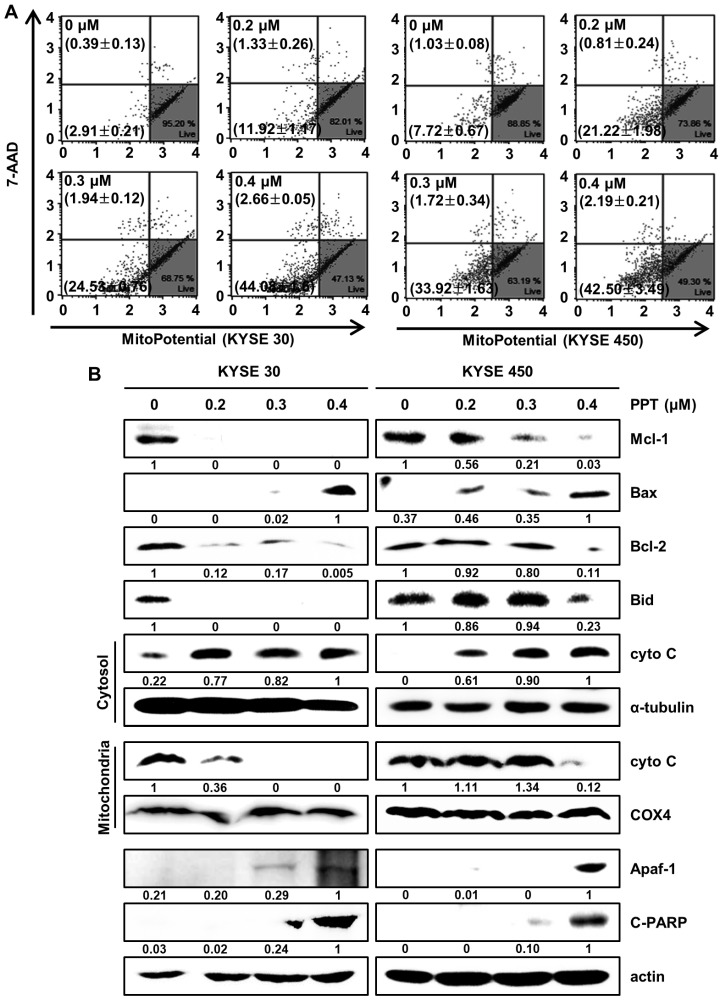
PPT alters mitochondrial membrane potential (MMP; Δψ) levels and induces apoptosis by regulating apoptosis-associated protein in ESCC cells. KYSE 30 and KYSE 450 were treated with PPT (0, 0.2, 0.3, and 0.4 µM) for 48 h. (**A**) The Muse™ MitoPotential Reagent (Merck Millipore) dye-stained cells were analyzed for the loss of MMP using a Muse™ Cell Analyzer (Merck Millipore). (**B**) The total proteins were quantitated and the expression of Mcl-1, Bax, Bcl2, Bid, cyto C (cytosol and mitochondria), α-tubulin, COX4, Apaf-1, and C-PARP detected by Western blots. The expression of actin was employed as an internal control.

**Figure 6 ijms-21-04640-f006:**
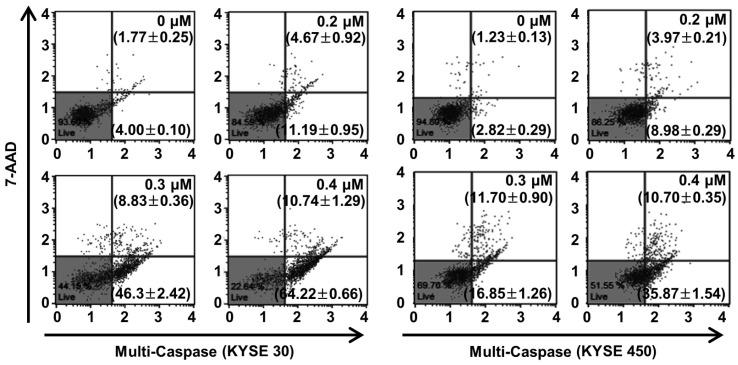
PPT treatment enhanced the multi-caspase activity in ESCC cells. The cells were treated with the indicated concentrations of PPT (0.2, 0.3, and 0.4 µM) or DMSO. The activation of multi-caspases was measured using a Muse™ Multi-Caspase Kit (Merck Millipore) and a Muse™ Cell Analyzer (Merck Millipore). The multi-caspase assays were conducted in triplicate.

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
