# Peer review of "Picropodophyllotoxin, an Epimer of Podophyllotoxin, Causes Apoptosis of Human Esophageal Squamous Cell Carcinoma Cells Through ROS-Mediated JNK/P38 MAPK Pathways"

_ijms, 2020, doi:10.3390/ijms21134640_

Round 1

Reviewer 1 Report

The paper by Kwak et al is an examination of picropophyllotoxin activity against esophageal squamous cell carcinoma. This is a form of cancer that has no current adequate treatment. The toxin was found to induce oxidative stress and apoptosis in human esophageal squamous cell carcinoma cells in culture. The study is a step in the right direction. It should be accepted in its current form.

Author Response

˜˜ Reviewer #1 Question:

Reviewer #1: The paper by Kwak et al is an examination of picropophyllotoxin activity against esophageal squamous cell carcinoma. This is a form of cancer that has no current adequate treatment. The toxin was found to induce oxidative stress and apoptosis in human esophageal squamous cell carcinoma cells in culture. The study is a step in the right direction. It should be accepted in its current form.

â–¶ Response to Question:

- Thank you for your compliments.

Reviewer 2 Report

The presented manuscript is a nice examplke of a thorough onco-pharmacological assessment of a natural compound with potential antineoplastic activities. Nevertheless some issues need to be rsolved in oreder to make it suitable for publication.

There is a MAJOR typographical error in the very title of the paper: 

Picropophyllotoxin, an epimer of podophyllotoxin, causes apoptosis of human esophageal squamous cell carcinoma cells through ROS-mediated JNK/p38 4 MAPK pathways

(it shoul be replaced with picropodophyllotoxin)

At row 188 the authors claim: "Although treatment with PPT is suitable for cancer chemoprevention, the mechanism is, as yet, not clear."

The whole study is based on end-points and bioassays indicative for chemotherapeutic potential, with inhibitory effects omn cancer cells. This has NOTHING to do with cancer prevention, whereby differfent processes take place and alternative compounds are investigated. Most of the usefull anticancer drugs and especially microtubule disrupting agents have absolutely no role in CANCER PREVENTION, indeed they could exert carcinogenic properties in paralel to most of the clinically useful classical anticancer drugs. Hence the notion that PPT is suitable for cancer chemoprevention is not scientifically justifiable and should be ommitted.  

Author Response

˜

˜ Reviewer #2 Question:

Reviewer #2: The presented manuscript is a nice example of a thorough onco-pharmacological assessment of a natural compound with potential antineoplastic activities. Nevertheless some issues need to be rsolved in oreder to make it suitable for publication.

There is a MAJOR typographical error in the very title of the paper: 

Picropophyllotoxin, an epimer of podophyllotoxin, causes apoptosis of human esophageal squamous cell carcinoma cells through ROS-mediated JNK/p38 4 MAPK pathways

(it shoul be replaced with picropodophyllotoxin)

At row 188 the authors claim: "Although treatment with PPT is suitable for cancer chemoprevention, the mechanism is, as yet, not clear."

The whole study is based on end-points and bioassays indicative for chemotherapeutic potential, with inhibitory effects omn cancer cells. This has NOTHING to do with cancer prevention, whereby differfent processes take place and alternative compounds are investigated. Most of the usefull anticancer drugs and especially microtubule disrupting agents have absolutely no role in CANCER PREVENTION, indeed they could exert carcinogenic properties in paralel to most of the clinically useful classical anticancer drugs. Hence the notion that PPT is suitable for cancer chemoprevention is not scientifically justifiable and should be ommitted.  

â–¶ Response to Question:

  • We have changed line 2. Also we have added line 187 and changed line 187,188, 192, 229-233.
  • Line 2: Picropophyllotoxin, an epimer of podophyllotoxin, causes apoptosis of human esophageal squamous cell carcinoma cells through ROS-mediated JNK/p38 MAPK pathways → Picropodophyllotoxin, an epimer of podophyllotoxin, causes apoptosis of human esophageal squamous cell carcinoma cells through ROS-mediated JNK/p38 MAPK pathways
  • Line 187: PPT is one of well-known naturally occurring aryltetralin lignans revealing anti-tumor activity [30].
  • Lind 188: Although treatment with PPT is suitable for cancer chemoprevention → Although PPT has indicated anti-tumor effects in several cancers [11, 12]
  • Line 192: The present study used a cellular model in which apoptosis was analyzed to examine the chemopreventive/therapeutic effects of PPT on ESCC esophageal cancer growth. → The present study used a cellular model in which apoptosis was analyzed to examine the anti-cancer effects of PPT on ESCC esophageal cancer growth.
  • Line 229-233: Combination therapy, which combines two or more treatments, is currently a central component of cancer therapy [41]. Fusion of anti-cancer drugs ameliorates efficacy compared to mono-therapy and potentially decreases drug resistance, offering therapeutic anti-cancer benefits such as reducing tumor growth and metastatic potential, cell cycle arrest, and induction of apoptosis [41, 42].
  • Line233-235: The findings of this study suggest that PPT may be valuable for cancer prevention and as a potential anticancer agent for human ESCC. → The findings of this study suggest that PPT may be valuable as a potential anti-cancer agent and be considered valuable candidates of combination therapy for human ESCC.
